# Probable Pain on the Pain Assessment in Impaired Cognition (PAIC15) Instrument: Assessing Sensitivity and Specificity of Cut-Offs against Three Standards

**DOI:** 10.3390/brainsci11070869

**Published:** 2021-06-29

**Authors:** Jenny T. van der Steen, Andrew Westzaan, Kimberley Hanemaayer, Muhamad Muhamad, Margot W. M. de Waal, Wilco P. Achterberg

**Affiliations:** 1Department of Public Health and Primary Care, Leiden University Medical Center (LUMC), P.O. Box 9600, 2300 RC Leiden, The Netherlands; awestzaan@hotmail.com (A.W.); k.b.m.hanemaayer@umail.leidenuniv.nl (K.H.); m.muhamad@svrz.nl (M.M.); M.W.M.de_Waal@lumc.nl (M.W.M.d.W.); W.P.Achterberg@lumc.nl (W.P.A.); 2Department of Primary and Community Care, Radboud University Medical Center, P.O. Box 9101, 6500 HB Nijmegen, The Netherlands; 3Aafje Thuiszorg, Huizen en Zorghotels, Pietersdijk 60, 3079 TD Rotterdam, The Netherlands; 4Franciscus Vlietland Hospital, Vlietlandplein 2, 3118 JH Schiedam, The Netherlands; 5Stichting Voor Regionale Zorgverlening (Nursing Home), Prins Clauslaan 1, 4691 ZA Tholen, The Netherlands

**Keywords:** dementia, pain measurement, behavior observation techniques, diagnostic techniques and procedures

## Abstract

Observational pain scales can help to identify pain in persons with dementia who may have difficulty expressing pain verbally. The Pain Assessment in Impaired Cognition-15 (PAIC15) covers 15 items that indicate pain, but it is unclear how probable pain is, for each summed score (range 0–45). We aimed to determine sensitivity and specificity of cut-offs for probable pain on the PAIC15 against three standards: (1) self-report when able, (2) the established Pain Assessment in Advanced Dementia (PAINAD) cut-off of 2, and (3) observer’s overall estimate based on a series of systematic observations. We used data of 238 nursing home residents with dementia who were observed by their physician in training or nursing staff in the context of an evidence-based medicine (EBM) training study, with re-assessment after 2 months in 137 residents. The area under the ROC curve was excellent against the PAINAD cut-off (≥0.8) but acceptable or less than acceptable for the other two standards. Across standards and criteria for optimal sensitivity and specificity, PAIC15 scores of 3 and higher represent possible pain for screening in practice, with sensitivity and specificity against self-report in the 0.5 to 0.7 range. While sensitivity for screening in practice may be too low, a cut-off of 4 is reasonable to indicate probable pain in research.

## 1. Introduction

Due to frequent co-morbid conditions that involve pain, pain is highly prevalent in persons with dementia and with other diseases that cause cognitive impairment or difficulty expressing pain, such as intellectual disability [1,2,3,4]. Various dementia types may differently affect the way pain is being perceived (threshold and intensity), processed, and communicated, but typically about half of persons with dementia are in pain [1,5,6]. Pain may be communicated non-verbally in manners that others may not always understand, for example, expressed as agitation. Indeed, better pain management in the context of a pain or needs assessment can effectively address challenging behavior [7,8]. However, longitudinal studies found associations between pain and challenging behavior to be weak with diverting patterns towards the end of life (increased pain, in particular at rest, whereas agitation decreased, e.g., [9,10]). These studies point to a need for cautious, evidence-based interpretation of possible behavioral indicators of pain.

The research community has been highly responsive to the problem of possible misclassification and under detection of pain, and by 2014, 28 pain observation scales had already been developed to improve detection of pain in these populations [11,12], while more measures have been developed more recently (e.g., Ersek et al. [13] and Richey et al. [14]). Many scales, however, have been developed based on clinical experience and expert opinion only, or were based on confounding with the concept of observable discomfort without any causal claim [15]. In response to a clear need for more systematic development of pain scales to optimize item pools, the Pain Assessment in Impaired Cognition-15 (PAIC15) was developed as a meta-tool [16]. Studies on psychometric properties in several countries and populations and refinement led to a scale with the reduced number of 15 items that represented the optimal item pool [17,18].

Use of an observational pain scale will only decrease pain if possible barriers to pain assessments are being addressed, including concerns for overmedication and lack of physician involvement [19]. Nursing staff reporting of pain is heavily impacted by relationships with other nurses and physicians [20]. Therefore, a clear multidisciplinary plan of action is needed which may include a basic agreement on *when* nursing staff should communicate pain observation scores to physicians. For this, staff should have confidence in a cut-off score for probable pain [21,22]. 

The exact cut-off may depend on routine use as a screening tool (high sensitivity preferred) or evaluation of effect of interventions (more balanced sensitivity–specificity for probable pain). Cut-off scores, for example, for the Pain Assessment in Advanced Dementia (PAINAD) [23], of 2 and higher to represent probable pain [24] can also be useful in protocols for trials (e.g., van der Maaden et al. [25]) because it offers actionable guidance. Further, cut-off scores can support tabulation of subgroups with and without pain (e.g., [10]), which facilitates reporting and interpretation of research findings. The goal of our study was to determine cut-offs for probable pain on the PAIC15 for use in practice and research determined based on sensitivity and specificity criteria against multiple available standards.

## 2. Materials and Methods

### 2.1. PAIC15

The PAIC was developed with a team of researchers supported by a European COST Action (TD-1005 “Pain assessment in patients with impaired cognition, especially dementia” [16]). Development procedures were thorough, with items abstracted from the main available pain scales, transparent item reduction procedures to retain the optimal combination of items, and testing of psychometric properties of items in parallel in more countries [17,18]. The resulting PAIC15 (also included in Appendix A) contains three of the six groupings of “common pain behaviors” in cognitively impaired older persons proposed by the American Geriatrics Society (AGS) [26]: “facial expressions,” “verbalizations, vocalizations,” and “body movements.” These three groups can be observed directly, while the other three AGS groups refer to behavioral changes observed over time.

### 2.2. Data Collection

#### 2.2.1. Data Source

We used data from an observational study on pain and discomfort in nursing home residents with impaired cognition. Resident-level data were collected in the context of mandatory evidence-based medicine (EBM) courses [27,28] that are part of a three-year residency internship of the elderly care physician training program [29] at Leiden University Medical Center (LUMC). Since 2018, physicians in training collect data on pain and discomfort in nursing home residents with cognitive impairment under their care. For the EBM training study, they attend classes on how to collect the data, enter data in an online Castor EDC (Amsterdam, The Netherlands) module, develop a proper research question, analyze the data, and report and present the results. The consecutive sample expands as the physicians in training each collect data on 5 nursing home residents they care for in homes mostly in the west and south of the country. Repeated assessments about 2 months apart allow for assessing changes over time. We used data collected by four trainee cohorts that started bi-annually with 250 assessments conducted between 26 September 2018 and 30 June 2020, selecting residents with a diagnosis of dementia.

#### 2.2.2. Reporting and Ethics

In the reporting of methods and results, we adhered to the Standards for Reporting Diagnostic Accuracy (STARD) statement list of essential items for reporting [30]. The key linking name to research-ID remained in the nursing home. The Medical Ethics Review Committee of the LUMC reviewed the protocol (number P18.100, 24 September 2018) and issued a waiver from the Medical Research Involving Human Subjects Act (WMO) review.

#### 2.2.3. Subjects

Data were collected mostly on psychogeriatric units where most physicians in training practiced during the EBM training study. These are closed departments with 24/7 oversight for, almost exclusively, persons with a physician diagnosis of dementia who typically stay for the rest of their life—in contrast to Dementia Special Care Units in the US designed for those with no severe dementia to improve functioning with behavioral problems or wandering [31]. Residents included in the study have a physician diagnosis of dementia or MCI or CVA with a Global Deterioration Scale (GDS) [32] score of 4 or higher. Residents with a life expectancy of shorter than one week were excluded. The physicians listed eligible residents by alphabetical order of surname and recruited residents until informed consent was provided for the number of residents aimed for. They asked the resident if capable, or the family, or both.

#### 2.2.4. Assessments

The first cohort comprised a single assessment of about 10 residents by the physician in training; for the next cohorts, the physicians in training were instructed to assess 5 residents twice. The second assessment of the fourth cohort, scheduled in spring 2020, lacked the direct observations of pain and discomfort when physicians were not allowed access to the residents for research purposes due to COVID-19 visiting restrictions. 

#### 2.2.5. Observations and Assessment Order

Data on pain and discomfort were collected through direct observation. The observations started with the physician’s or instructed staff member’s assessment of (1) Pain Assessment in Advanced Dementia (PAINAD) [23,24], 2 min (summed score and cut-off not shown on the form), (2) PAIC15, 3 min, (3) Discomfort Scale—Dementia of Alzheimer Type (DS-DAT) [15,33,34], 5 min, (4) self-report of being in pain (yes/no) and pain intensity on a 0 to 10 scale for two moments, now (used in the analyses) and last week, (5) observer’s overall estimates of being in pain (yes/no) and pain intensity integrating observations and self-reported pain, and finally, (6) an open-ended item on any problems or circumstances encountered during observations. 

The physicians were trained in assessing the observational DS-DAT as the most complicated instrument with an instructional video and filmed patients (no actors) rated by the scale’s developer. The 9 DS-DAT items are scored 0–3 and include detailed descriptors of observable behavior and consider frequency, intensity, and duration of the behavior, and therefore prompts trainees to accurate and precise observations, without interpreting behavior with respect to possible sources [27,28]. The physicians could delegate the observations if they instructed other staff how to conduct the observations.

#### 2.2.6. Data Processing

The physicians entered data in Castor EDC directly, or after first completing a printed questionnaire of the data collection tool. The accuracy of data collection was verified by independent re-entry of quantitative data of 19 of 99 available print questionnaires: 11 first assessments selecting the first entry and 8 second assessments selecting the last questionnaire entered by the physician. Inaccuracies were identified in 0.41% of data items (a total of 21 inaccuracies in 19 questionnaires each with 270 data items). Inaccuracies included minor problems such as observation times that were minutes off—all were corrected.

### 2.3. Measures and Pain Standards

To describe the sample, we report demographics, type of dementia, dementia stage with the GDS [32] and the Bedford Alzheimer Nursing Severity-Scale (BANS-S) [35,36], full ADL dependency with BANS-S items, any acute disease at the time of the assessment, and presence of comorbid disease that may be related to pain from the categories of the Functional Comorbidity Index (FCI) [37,38]. A total score ranging 0 to 36 is calculated summing 18 items, coded 0 if not present or present but no influence on daily functioning, score 1 if the comorbid condition is partially of influence, and 2 for severe influence. 

The PAIC15 comprises 15 items, all negatively phrased indicators, with brief item descriptors (Appendix A). Response options include “not at all” (with 0 points shown on scoring form), “slight degree” (1 point), “moderate degree” (2 points), “great degree” (3 points), and “not scorable.” Weight of the 15 items is equal, so summed scores total 0 to 45.

To arrive at a PAIC15 cut-off, we used three standards of probable pain: (1) self-report when able: asking the resident about pain and intensity, (2) direct observations with the PAINAD using cut-off 2 [24], and (3) overall estimation of the physician or nurse after the observations and self-report. Although we expected high sensitivity and specificity against the PAINAD cut-off because of overlap with PAIC15 items, we used self-report of pain at the time of the observations, if able, as the generally endorsed preferred standard, which should be attempted despite limitations [5,39] to assess a cut-off on the PAIC15. 

To support self-report when asking about pain [40], after the observations to assess the PAINAD and PAIC and asking if the resident was in pain, the resident was shown a pain intensity scale if they indicated to be in pain (Appendix A). The pain intensity scale was an enhanced Visual Analog Scale (VAS) horizontal numerical scale with the colors and the verbal descriptors used with MDS3.0 [41]. Unable to respond was an explicit option. 

We used the established 0-10 PAINAD scale [23] with 5 items with verbal descriptors of 3 response options, which can be completed reliably by staff unfamiliar with the person in 1 to 5 min [42]. An attempt to console was part of the PAINAD assessment procedure when the observer feels there may be a need to console. A cut-off of 2 was based on multiple sources—using two datasets and a Doloplus cut-off [24]. PAINAD is, along with PACSLAC, one of the most used, translated, best-tested, and often recommended pain observation scales for persons with dementia [11,22,43], each having particular weak points (e.g., PAINAD for breathing item reliability problems recognized since its development [23], and PACSLAC for facial expression reliability [44] and length [42] and lack of descriptors [15]). Although a cut-off is also available for the PACSLAC [24], the PACSLAC was not developed for use with a cut-off. The developers recommend an individualized approach comparing scores over time for use in practice [45]. 

Finally, following observations for the PAINAD and PAIC, and self-report, the observer’s overall estimate of being in pain (yes/no) and if yes, pain intensity with a 0 to 10 score, was assessed after the observations, in the same way as self-reported pain (Appendix A). Therefore, at that point, the observer could integrate all information available from knowing the resident, the observations, and self-reported pain.

### 2.4. Analyses and Cut-Off Assessment 

Since the distributions of all pain scores were skewed, we calculated Spearman correlations of pain scores at assessment 1 and 2, and of the PAIC15 score with the three standards. We also calculated correlation with the DS-DAT as a related, overlapping, yet different concept [15], completed after PAINAD and PAIC15 to explore construct validity, expecting correlations of 0.30 to 0.50. Between the same constructs (PAIC15 and the three pain standards), we expected correlations of ≥0.50 [46].

Receiver Operator Characteristic (ROC) curves with 95% confidence intervals were calculated, with plots showing sensitivity against 1 minus specificity. We calculated these separately for the two assessments because there are no accepted methods to account for clustering of assessments in ROC curves. Further, with the second assessment, the physician knew the resident better, which might affect the overall estimation of pain (first assessment: relatively new, second assessment, physician had known the resident for at least a couple of months). The area under an ROC curve of 0.5 (a 45-degree diagonal line) indicates that there is no discrimination, so in this case, patients with and without pain according to the standard cannot be separated. Hosmer and Lemeshow [47] regard areas from 0.7 to 0.8 as acceptable discrimination, from 0.8 to 0.9 as excellent, and values from 0.9 as outstanding. 

Criteria to select a cut-off were three-fold: (1) cut-off with the highest summed sensitivity and specificity (i.e., the value on the ROC curve that is at the upper left corner), (2) at the highest sum, in particular for use in practice, sensitivity should be at least as high as specificity, and if sensitivity is lower than specificity at the highest sum, we choose the cut-off for which sensitivity and specificity are most balanced (closest to each other while still requiring sensitivity equal or higher than specificity), and (3) if findings differed across the three standards, we prioritized the cut-off determined against self-report.

We adhered to a level of significance of *p* < 0.05 for all analyses. To not drop cases for just missing a single or a few scale items, as done previously [48], we imputed missing item scores with the resident mean item score if at least two-thirds of the items was available (which implies maximum 5 imputed items on the PAIC15, 1 on the PAINAD, 3 on the DS-DAT, 2 on the BANS-S, and 6 on the FCI). 

We regarded “not scorable” on the PAIC15 and the DS-DAT as missing values subject to the imputation procedure. However, considering ease of use in practice, we additionally conducted sensitivity analyses, first repeating the analyses with total scores, ignoring the non-scorable missing items (in fact, imputing 0 to arrive at the total score, except for imputing the groaning item because it was observed but data entry failed in the first trainee cohort). Next, we ran the analyses, omitting an item that was often rated as non-scorable because it could not be observed (resisting care; in fact, using a 14-item version of the PAIC15), because in practice, this would facilitate calculating total scores. 

We also conducted sensitivity analyses choosing three different samples to check if this affected the cut-off. First, we reran the analyses with the two observer standards in the selected sample of residents able to self-report. We also reran the analyses without the cases in which the physician attempted to console the resident before the PAIC15 assessment as this would have provided observers with more information than commonly collected for the PAIC15. Finally, we omitted cases with missing values, conducting complete case analysis. We present classifications against the chosen standards by cross-tabulating the cut-off against the three standards with two-by-two tables to include these raw counts, as recommended in the STARD statement [30]. Additionally, guided by STARD, we verified comments with the open-ended item on circumstances for any adverse events as a result of the pain assessments of the PAIC15 and three standard instruments. Analyses were conducted with SPSS 25 (IBM, 2017).

## 3. Results

Characteristics of the selected 238 residents with dementia at the first assessment and 137 of these residents at the second assessment (Figure 1, flow chart) are shown in Table 1. The second assessment was not conducted (Figure 1) when it was not part of the training course (first trainee cohort), when unable to visit the resident in response to COVID-19 measures, and when the resident had died. The average time between assessments for the residents with two assessments was 2.10 months (SD 0.41). The data were collected by 44 physicians in training in 37 nursing home facilities (each facility hosting up to three trainees consecutively, and one physician collected data in two facilities). The second assessment was conducted by 29 physicians in 26 facilities. Per facility, data were collected for 1 to 20 residents at the first assessment, and 3 to 11 residents at the second assessment. Of the pain and discomfort observations, most (79.4% at assessment 1 and 77.4% at assessment 2) were conducted by the physicians themselves, and 7.1% and 10.2% respectively, by other staff, mostly nurses, and the remaining 13.4% and 12.4% respectively, by physicians and other staff together or in part by each.

The residents comprised a nursing home population with about two-thirds female residents and mean ages around 86 years (Table 1). About two-thirds had dementia with an Alzheimer’s component, and one-third a vascular component. For about one-third, the dementia was in a severe stage, consistent with less than half being fully dependent in the main ADL functions. Potentially painful conditions were prevalent.

Mean DS-DAT discomfort scores were 5.6 at the first assessment (SD 5.2) and 5.3 at the second assessment (SD 5.0). Mean PAIC15 scores (Table 2) were 4.6 at the first assessment and 4.0 at the second assessment (SD 5.2 for both assessments). For the PAINAD too, the SD was greater than the mean (mean 1.0, SD 1.6 for both assessments), indicating skewed distributions, and no resident reached the theoretical maximum of the total scores (Table 3 and Table 4). The distributions of self-report pain intensity and observer’s overall estimate on the 0–10 scales were skewed too, with means below 1, SD between 1 and 2, and a rating of 10 occurring just once (Table 2, Table 3 and Table 4). At the first assessment, 197 of 238 (82.8%) residents were able to self-report any pain, and 177 (74.4%) could provide a pain score. For the second assessments, these figures were 110 (80.3%) and 99 (72.3%) of 137, respectively. Of the residents who were in pain and provided a pain score (27 at the first assessment, 10 at the second assessment), the means were 4.89 (SD 2.12) and 4.60 (SD 1.58), respectively.

Table 5 shows that the correlation of the two PAIC15 assessments with the pain instruments used as pain standards was highest for the PAINAD (0.72 at both assessments), which was slightly higher than for the DS-DAT (0.69 and 0.63, respectively). It was lowest for self-report (0.09 and 0.06, no significant correlation). The correlation with the observer’s overall estimate rated in-between, with a lower correlation at the first assessment (0.28) compared with the second assessment (0.41). Correlations between two assessments with the same instrument ranged from 0.47 to 0.57, including for self-report, except for the observer’s overall estimate, with a somewhat lower correlation of 0.32.

Table 6 shows that discrimination was consistently good for both assessments against the cut-off for the PAINAD only (“excellent” with area under the ROC curve 0.87 and 0.88 and also the lower CI was ≥0.8). Discrimination was acceptable for the second observer’s overall estimate (0.78), almost acceptable for the first one and the second self-report (0.69), and inadequate against the first self-report (0.58).

Table 7 and the ROC in Appendix A show that against self-report, the highest summed sensitivity and specificity with the highest sensitivity (marked green) were for cut-offs of 2.7 and 2.6 at assessment 1 and 2, respectively. Appendix A shows that for the PAINAD, this was reached with cut-offs at 3.7 and 2.6 respectively, and against the observer’s estimate, at 2.7 and 3.6. For a rounded cut-off of 3, sensitivity and specificity against self-report would be in the 0.5 to 0.7 range, with sensitivity against the other standards up to 0.9.

Most PAIC15 items did not have non-scorable or otherwise missing items, or at most 2 missing values. However, the item “resisting care” was often not rated because observations were usually not during caregiving (assessment 1: 41 non-scorable, 17.2%; assessment 2: 25 non-scorable, 18.2%; Appendix A). Compared with imputed items, when imputing missing values with 0, the mean PAIC15 item score was 0.1 lower, with 4.5 (SD 5.0) at assessment 1, and at assessment 2 (3.9, SD 5.0). Attempts to console the resident were undertaken in 40 cases at assessment 1, and in 20 cases at assessment 2, before the PAIC15 assessment, as instructed, with the PAINAD if needed.

Across all sensitivity analyses, areas under the ROC curve were mostly similar, with small changes (<0.05) in different directions, except for a drop from 0.78 to 0.73 against the observer’s estimate at assessment 2 in the selection able to self-report, and a drop from 0.69 to 0.64 at assessment 1 for complete cases. In only one case, the difference crossed the acceptable criterion (0.69 against self-report at assessment 2 increased to 0.70 in the selection of cases with no need to console). All sensitivity analyses supported a cut-off of 3 or 4 against the PAINAD or the observer’s estimate, and a cut-off of 3 against self-report (cut-offs were often around 2.5). With a cut-off of 3, 30% of patients would self-report pain (assessment 1, 33/109; assessment 2, 15/50), while with PAIC15 scores lower than 3, this would be about half (10% at assessment 2 (6/60) to 17% (15/88) at assessment 1; Table 8). 

Nevertheless, the cross-tabulations in Table 8 illustrate that a cut-off of 3 already presents a conservative approach, with fewer cases of pain missed (false negatives) compared with unjustified identification of pain (false positives) according to the standards. Only the cut-off of 4 against the PAINAD would represent probable pain in the sense of pain in the majority of cases with PAIC15 scores of 4 or higher, and in the case of cut-off 3 for the second PAINAD assessment.

In a few cases, possible adverse events were reported, with the patient possibly becoming nervous or anxious being observed by a physician whom the patient did not know, or the patient had difficulty answering the question about pain. On the other hand, some patients were eager to connect with the physician and the physician identified unsafe situations and symptoms while visiting for the observation. 

## 4. Discussion

Against three available standards, the cut-off for pain on the PAIC15 is 3 when self-report is prioritized over structured observation and an observer estimate of pain, and sensitivity is prioritized over specificity. Scores of 3 or higher would flag possible pain upon screening, with sensitivity and specificity in the 0.5 to 0.7 range for self-report and sensitivity up to around 0.9 for the other standards. Further, scores of 4 and higher would represent probable pain on the PAIC15 against the other observational pain scale, the PAINAD, with the highest correlations (coefficient 0.72), which is the only standard against which areas under the ROC curve were excellent (≥0.8). Although the PAIC15 did not correlate with self-reported pain (coefficients < 0.10), the optimal cut-off against self-report (3) was close to those of the observational standards or one point lower (4), which suggests a risk of underestimating pain when fully relying on observations compared to self-report. More congruence (higher values of areas under the ROC curve and better correlation with the observer’s estimate) at the second assessment suggests that knowing the patient better may help in eliciting self-reported pain or better estimates of pain. Dealing with missing values is relevant for use in clinical practice. We found that summing items without the item resistance to care (leaving it blank), which cannot always be observed (e.g., when there is no caregiving), did not affect the recommended cut-offs for pain.

Over four of five residents (80–83%) could say whether they were in pain or not, and almost three quarters of the nursing home patients in our study were able to self-report on a combined numerical/verbal/color scale. This is in line with Pautex et al., who found that 61% of hospitalized patients with severe dementia was able to self-report pain on at least one of three self-report scales (verbal, visual, faces) [49]. Across studies in institutional settings, a VAS could be completed by about half to 90% of patients with dementia [42]. Lukas et al. [50], at a geriatric hospital, found that 90% of patients with moderate dementia (Mini-Mental State Examination, MMSE, scores 10–17) were able to complete 2 or 3 of 3 self-report scales, but of 17 patients with MMSE < 10, only 1 patient was able to do so (combined moderate and severe dementia, 68%, 39 of 57). In community settings, between 53% and 67% of persons with moderate to severe dementia were able to self-report (four studies [51]).

Although somewhat higher than in our study, Kwon et al. [42] also found low correlation (0.36) with self-report using a Numerical Rating Scale (NRS) verbalizing descriptors at rest, while correlations amounted to 0.76 to 0.81 with the Face, Legs, Activity, Cry, and Consolability (FLACC) observational scale. Lukas at al. [50] found low or negligible correlations between self-report and the observational scale (PAINAD) when observed at rest, but with movement, correlations amounted to around 0.5. Of concern, they found that four observational scales, including the PAINAD, did not differentiate between a decrease of pain in the control group versus an oxycodone-treated group with advanced dementia in geriatric hospitals [52]. This is, however, in contrast to the few other available responsiveness studies [53], in a small blinded trial with oxycodone. Responsiveness relates to intensity of pain, and Mosele et al. [54] found that intensity of pain at the NRS related to PAINAD scores across levels of cognitive impairment, which suggests that it may be possible to also assess cut-offs for intensity of pain, but findings in regard to linearity have been inconsistent [42]. Ersek et al. [13] developed a pain intensity measure, and they found that rating intensity or frequency of individual behaviors did not affect selection of items for the measure, which supports the use of pain observational scales to detect pain, pain intensity, and changes in pain intensity. 

A strength of our study is the use of trained observers who conducted direct observations, which excludes recall bias. A possible limitation is that we missed some potentially valid self-reported pain as we showed the self-report scales only if patients replied affirmative to the pain question. Further, we used a horizontal self-report scale, while for persons with a CVA (almost a quarter in our study), a vertical scale [41] might be preferred. However, Pautex et al. [40] found no difference in understanding of horizontal (81%) and vertical (77%) self-report scales, while more than half had dementia with a vascular component (16% vascular dementia and 37% mixed), which was higher than in our study (32–35%). We did not randomize the order of the PAINAD and PAIC15 observations. There might have been an effect on the PAIC15 assessment from additional intervening in an attempt to console as part of the last PAINAD item, consolability, but this was unlikely based on the sensitivity analyses.

That is, we ran a series of sensitivity analyses which underpinned the robustness of our findings. The extensive analyses also showed some variability even with assessment 2 months apart in the same residents and different cut-offs depending on preferred high sensitivity (for screening in practice) or balanced sensitivity and specificity (for research). Cut-offs may, however, also differ in populations with different levels of self-reported pain and for situations in which pain is being induced (e.g., during care activities or as part of the pain assessment). 

The correlations and areas under the ROC curves indicate that, as expected, the PAIC15 resembles the PAINAD—a related observational scale—closer than self-report or the observer’s estimate of pain. The fact that we cannot come up with a single best cut-off illustrates that there are still ambiguities around the assessment of pain in persons with dementia. This is pertinent to observational pain scales, yet there is a clear need to also assess pain in about a quarter to a third of patients in community and institutional settings of patients with moderate to severe dementia who are unable to self-report. Cautious interpretation of findings in practice and in research is warranted given our finding of considerable false positives and false negatives against self-report. Despite rather low and different specificity against different standards, cut-offs for possible or probable pain are useful as part of pain protocols as they can help trigger further identification of pain.

Future research may examine cut-offs in potentially painful situations, such as with movement or a flu injection, with more power to also examine possible cut-offs for severe pain. Further, there is a need to better distinguish non-pain discomfort behavior, as we found that correlation with the discomfort scale was higher than expected for related yet overlapping concepts [46], and only slightly lower than correlation with the other pain scale, PAINAD. Future research may separately consider situations in which the observer knows and does not know the patient, while optimal standards may be assessed in the first situation, but observational tools are most needed for the latter situation. It is also possible that knowing the patient results in possible familiar behavioral indicators being overlooked. Further research may evaluate cut-offs as part of pain protocols, which may even proactively include consideration of painful conditions and procedures [4]. In addition, other dimensions of pain need to be considered, such as chronicity and location [5].

## 5. Conclusions

Across two assessments and against three standards, including self-report, we found cut-offs at the PAIC15 of 3 and 4 to represent possible or probable pain for use in practice and research, respectively. Guided the most by self-report, for screening in practice, we recommend PAIC15 scores of 3 and higher to represent possible pain with sensitivity and specificity in the 0.5 to 0.7 range. For screening in practice, sensitivity of a cut-off of 4 may be lower than desirable, while for use in research with more balanced false positive and false negative counts, a cut-off of 4 would be a reasonable alternative.

## Figures and Tables

**Figure 1 brainsci-11-00869-f001:**
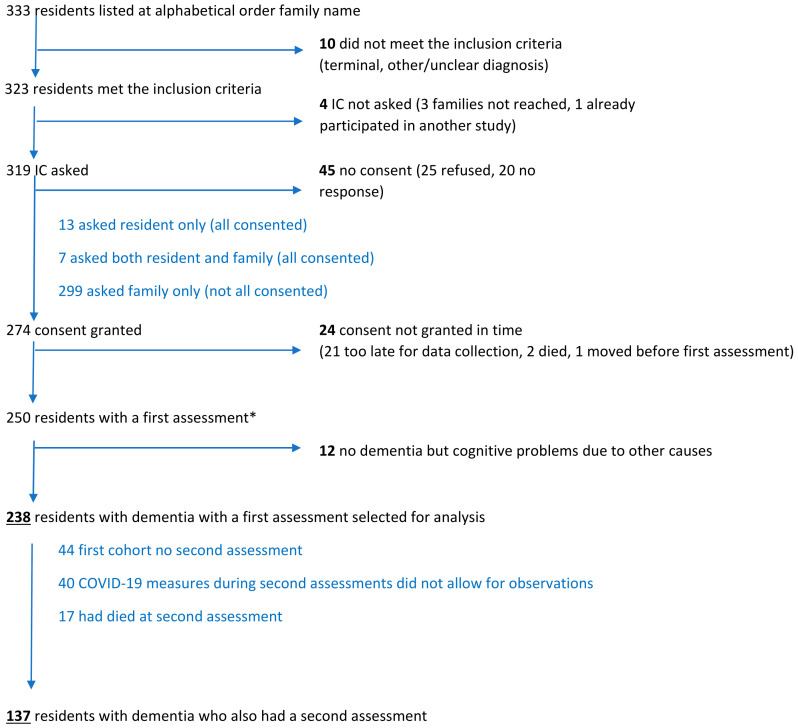
Flow chart of inclusion of residents. IC = informed consent.

**Table 1 brainsci-11-00869-t001:** Description of the sample of nursing home residents with dementia and pain.

	Assessment 1 (n = 238)	Assessment 2 (n = 137)
Women, % (n)	63.4% (151)	67.9% (93)
Age, mean (SD)	85.7 (7.5)	86.3 (7.8)
Residence, type of ward, % (n) psychogeriatric unit (almost all dementia) unit for mostly physical disability or combined physical and cognitive impairment	96.2 (229) 3.8 (9)	94.9 (130) 5.1 (7)
Type of dementia (more possible), % (n)		
Alzheimer’s	68.5 (163)	67.9 (93)
vascular	35.3 (84)	32.1 (44)
Lewy body or Parkinson	5.0 (12)	4.4 (6)
other types/mixtures and unknown	10.9 (26)	11.7 (16)
Stage of dementia		
BANS-S, mean (SD)	14.4 (4.5)	14.4 (4.9)
BANS-S 17 and higher, % (n)	33.6% (80)	33.6% (46)
GDS, mean (SD)	5.7 (0.86)	5.8 (0.84) ^1^
GDS 7, % (n)	16.0%(38)	19.1% (26) ^1^
ADL dependency, % (n)		
full dressing dependency	42.0 (100)	43.8 (60)
full mobility dependency	25.2 (60)	26.3 (36)
full eating dependency	6.3 (15)	7.3 (10)
dependency 0–6 scale, mean (SD)	2.7 (1.7)	2.7 (1.8)
Acute disease at time of assessment, % (n)	9.7% (23)	11.7% (16)
Weighted Functional Comorbidity Index, mean (SD)	8.5 (4.0)	8.8 (4.2)
Co-morbidity potentially related to pain, % (n)		
degenerative disc disease (e.g., back disease, spinal stenosis, or severe chronic back pain)	31.5 (75)	29.9 (41)
arthritis	25.2 (60)	27.8 (38)
cerebrovascular accident (CVA)	24.4 (58)	23.4 (32)
diabetes mellitus type I or II	22.3 (53)	28.5 (39)
depression	18.9 (45)	23.4 (32)
peripheral vascular disease	12.2 (29)	13.1 (18)
angina pectoris	11.3 (27)	10.9 (15)
DS-DAT (discomfort score mean (SD)/median (IQR))	5.6 (5.2)/4.5 (1–9) ^1^	5.3 (5.0)/4 (1–8)

BANS-S = Bedford Alzheimer Nursing Severity-Scale (scores of 17 and higher represent severe dementia) [35,36], GDS = Global Deterioration Scale, DS-DAT = Discomfort Scale—Dementia of Alzheimer Type. ^1^ One missing value.

**Table 2 brainsci-11-00869-t002:** Pain assessments mean, (SD)/median (IQR) (n).

Pain Assessment with Theoretical Range	Assessment 1	Assessment 2
PAIC15 (0–45)	4.6 (5.2)/3 (1–6) (238)	4.0 (5.2)/2.1 (0–5.5) (137)
Self-report intensity (0–10)	0.75 (1.94)/0 (0–0) (177)	0.46 (1.47)/0 (0-0) (99)
PAINAD (0–10)	1.1 (1.6)/0 (0–2) (238)	1.1 (1.6)/0 (0–2) (137)
Observer’s overall estimate (0–10)	0.81 (1.70)/0 (0–0) (237)	0.69 (1.56)/0 (0–0) (136)

PAIC15 = Pain Assessment in Impaired Cognition-15, PAINAD = Pain Assessment in Advanced Dementia (PAINAD).

**Table 3 brainsci-11-00869-t003:** PAIC15 assessment.

Score ^1^	Assessment 1 (n = 238)	Assessment 2 (n = 137)
0	16% (39)	28% (38)
1	12% (29)	12% (16)
2	13% (32)	14% (19)
3	13% (30)	7% (10)
4	10% (24)	12% (16)
5	6% (15)	3% (4)
6	6% (14)	4% (5)
7	1% (3)	3% (4)
8	5% (13)	3% (4)
9	3% (7)	5% (7)
10 to 20	11% (25)	9% (12)
20 and up	3% (7) (max score 34)	1% (2) (max score 33)

^1^ With imputation, rounded off to integers.

**Table 4 brainsci-11-00869-t004:** Pain standard assessments.

Measure (Scale or Assessment)	Score ^1^	Assessment 1 (n = 238)	Assessment 2 (n = 137)
Self-report, if able	**any pain** **(standard A)**	**23.9%** (n = 47/197)	**19.1%** (n = 21/110)
Self-report, intensity	0 no pain	(150)	(89)
	pain, but no score	(20)	(11)
	1 mild	(0)	(0)
	2 mild	(2)	(0)
	3 mild	(8)	(3)
	4 mild	(3)	(2)
	5 moderate to severe	(5)	(3)
	6 moderate to severe	(1)	(1)
	7 moderate to severe	(5)	(0)
	8 moderate to severe	(2)	(1)
	9 moderate to severe	(0)	(0)
	10 very severe/horrible pain	(1)	(0)
PAINAD	0	51% (121)	52% (71)
	1	21% (50)	21% (29)
	**2 and up** **(standard B)**	**28.2% **(67) (max score 8)	**27.0% **(37) (max score 8)
Observer’s overall estimate	**any pain** **(standard C)**	**23.2%** (55; 1 missing)	**19.7%** (27/137) ^2^
Observer’s overall estimate, intensity	0	(182)	(110) ^2^
	1	(2)	(2)
	2	(17)	(4)
	3	(15)	(5)
	4	(10)	(10)
	5	(5)	(2)
	6	(2)	(2)
	7	(1)	(1)
	8	(2)	(0)
	9	(0)	(0)
	10	(1)	(0)

^1^ With imputation, rounded off to integers. ^2^ Of 27 in pain, 1 missing intensity.

**Table 5 brainsci-11-00869-t005:** Correlations (Spearman) of two assessments with, on average, 2.6 months in-between and correlations of PAIC15 with the pain standards and discomfort.

	Pain				Discomfort
	PAIC15	Self-Report (Standard 1)	PAINAD (Standard 2)	Observer’s Overall Estimate (Standard 3)	DS-DAT
PAIC with other instruments: assessment 1		0.09	0.72 *	0.28 *	0.69 *
PAIC with other instruments: assessment 2		0.06	0.72 *	0.41 *	0.63 *
Same instrument: assessment 1 withassessment 2	0.57 *	0.47 *	0.55 *	0.32 *	0.52 *

* *p* < 0.05. PAIC15 = Pain Assessment in Impaired Cognition-15, PAINAD = Pain Assessment in Advanced Dementia, DS-DAT = Discomfort Scale—Dementia of Alzheimer’s Type.

**Table 6 brainsci-11-00869-t006:** Area under ROC curve.

Standard	Assessment 1 Area (CI)	n	Assessment 2 Area (CI)	n
Against self-report	0.58 (0.49–0.67)	197	0.69 (0.56–0.81)	110
Against PAINAD cut-off 2	0.87 (0.82–0.92)	238	0.88 (0.83–0.94)	137
Against observer’s overall estimate	0.69 (0.61–0.76)	237	0.78 (0.68–0.89)	137

**Table 7 brainsci-11-00869-t007:** Sensitivity and specificity of PAIC15 cut-offs against self-report.

Assessment 1			Assessment 2		
PAIC15 Score ^1^	Self-Report ^2^		PAIC15 Score ^1^	Self-Report ^2^	
	Sensitivity	Specificity		Sensitivity	Specificity
0.5	0.896	0.188	0.5	0.857	0.326
1.0	0.813	0.309	1.0	0.810	0.416
1.5	0.771	0.315	1.5	0.810	0.438
2.1	0.688	0.456	2.1	0.714	0.551
2.2	0.688	0.470	2.6	0.714	0.607
2.7	0.688	0.490	3.1	0.619	0.674
3.1	0.521	0.577	3.6	0.619	0.685
3.3	0.521	0.611	4.1	0.476	0.787
3.7	0.521	0.617	4.6	0.429	0.809
4.1	0.375	0.678	5.5	0.381	0.843
4.5	0.375	0.698	6.5	0.333	0.865
4.8	0.375	0.705	7.3	0.286	0.888
5.2	0.354	0.765	7.8	0.286	0.899
5.6	0.313	0.765	8.5	0.238	0.910
5.9	0.313	0.772	9.3	0.143	0.955
6.2	0.188	0.792	10.8	0.143	0.966
6.7	0.167	0.799			
7.3	0.167	0.819			
7.8	0.167	0.826			
8.0	0.146	0.879			
8.5	0.146	0.886			
9.1	0.146	0.906			
9.6	0.125	0.906			
10.4	0.125	0.933			

^1^ Imputed PAIC15 values shown up to 10 and the values represent the coordinates of the ROC curves in the Appendix A. ^2^ Pain and intensity 1 and up versus others for those who reported. In green: highest sensitivity plus specificity (coordinate of the ROC-curve with the most upper left position). In blue: sensitivity and specificity most balanced (values closest) for this cut-off. Red font: sensitivity lower than specificity.

**Table 8 brainsci-11-00869-t008:** Cross-tabulations of two PAIC15 cut-offs at two assessments by the three standards (counts).

a. Against self-report					
Assessment 1	Reported no pain	Reported pain	Assessment 2	Reported no pain	Reported pain
PAIC15 < 3	73	15	PAIC15 < 3	54	6
PAIC15 ≥ 3	76	33	PAIC15 ≥ 3	35	15
PAIC15 < 4	92	23	PAIC15 < 4	61	8
PAIC15 ≥ 4	57	25	PAIC15 ≥ 4	28	13
b. Against PAINAD					
Assessment 1	No pain observed	Pain observed	Assessment 2	No pain observed	Pain observed
PAIC15 < 3	94	6	PAIC15 < 3	72	1
PAIC15 ≥ 3	77	61	PAIC15 ≥ 3	28	36
PAIC15 < 4	121	9	PAIC15 < 4	77	6
PAIC15 ≥ 4	50	58	PAIC15 ≥ 4	23	31
c. Against observer’s estimate					
Assessment 1	No pain observed	Pain observed	Assessment 2	No pain observed	Pain observed
PAIC15 < 3	91	8	PAIC15 < 3	67	6
PAIC15 ≥ 3	91	47	PAIC15 ≥ 3	43	21
PAIC15 < 4	111	18	PAIC15 < 4	77	6
PAIC15 ≥ 4	71	37	PAIC15 ≥ 4	33	21

## Data Availability

The data presented in this study are available from the corresponding author upon reasonable request. The data are not publicly available because informed consent applied to use of the data for the purposes of answering questions in the evidence-based medicine training study, not necessarily for other purposes, for which consent of residents and families was an additional option.

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
