# Peer review of "Probable Pain on the Pain Assessment in Impaired Cognition (PAIC15) Instrument: Assessing Sensitivity and Specificity of Cut-Offs against Three Standards"

_brainsci, 2021, doi:10.3390/brainsci11070869_

Round 1
Reviewer 1 Report
I read this paper with interest. The paper has great potential to contribute to the literature on the topic of pain assessment in older adults with dementia. However, as written, the manuscript has many limitations. My comments follow:
- The fit between the manuscript and this journal is not obvious.
- The manuscript is very challenging to read. It would greatly benefit from better organization and substantial editing for flow, as well as the inclusion of subheadings.
- The last two paragraphs of the introduction (page 2, lines 57-73) don’t support the purpose of this paper. The sentences starting with “nursing staff reporting….” (lines 59-62) is irrelevant to determining a cut-off score on a new measure.
- Methods:
- In the data collection process, was the order of the measures randomly varied to reduce order effects? Why or why not?
- What was the rationale for including patients with CVA in the sample?
- The paragraph on page 3, lines 122-136, is very difficult to follow.
- This section would greatly benefit from subheadings - - sample, measures, data collection process, data collectors, and reference measures are not clearly articulated.
- What was the rationale for including “observer’s overall estimate” of pain?
- Details of the analysis section could be more clearly presented. Missing data procedures are not clear.
- Tables do not fit on the page and thus, are very difficult to follow. In Table 2, what is the rationale for considering self-reported pain scores of 5 – 9 (on 0-10 scale) as moderate to severe? Is there a reference for this? Why was the category “pain but no score” including in this table and not counted as missing data? Table 5 is unreadable.
- The results are not clearly presented. Therefore, it is not possible to evaluate the conclusions drawn in the discussion section. As presented, the cut-off score of 3 or 4 is not very useful.
Author Response
refer to uploaded response

Reviewer 2 Report
The Ms brainsci-1246172 entitled “Probable pain on the Pain Assessment in Impaired Cognition 2 (PAIC15) instrument: Assessing sensitivity and specificity of 3 cut offs against three standards” by van der Steen et al. reports on results from an observational study on pain and discomfort in nursing home residents with impaired cognition concerned with sensitivity and specificity of cut offs for probable pain on the Pain Assessment in Impaired Cognition-15 (PAIC15) tool against three standards. This research fills a gap and I want to congratulate with the Authors for this work. The novelty of this work consists in the evaluation of 3 cut offs for probable pain on PAIC15 against three different standards of evaluation and it is a fundamental process needed for the best use of this pain assessment tool.
The title and the keywords represent exactly the whole work.
The intoductory paragraph explores very carefully the panorama of pain in the context of impaired cognition, dementia particularly, pointing at the debated meaning of the behavioral indicators of pain and explaining the rationale underlying the PAIC15. In fact, the residents included in the study have a physician diagnosis of dementia or MCI or CVA. The growing attention to the problem of misdiagnosis and underdetection of pain has led to a proliferation of pain scales and the Authors highlight the need for systematic multidisciplinary development of pain scales instead of tools based on expert opinion only. This is a fundamental process for the definition of a definite cut off for probable pain and a consequent improvement of pain management in this fragile population in clinical practice. Also, the role of cut offs in research and the difference between use of the scale as screening tool or for evaluation of the efficacy of interventions is explained very clearly. Therefore, the Introduction provides the reader with all the necessary information for the contextualization of the research work reported in the Ms.
The methods used are described in detail, also with regard to COVID-19 restrictions, and the protocol results very well planned. The description of the PAIC-15 including groups of common pain behaviors (AGS) that can be observed directly underlines the feasibility of this meta-tool. Moreover, the Standards for Reporting 91 Diagnostic Accuracy (STARD) statement list of essential items for reporting were used for the reporting of methods and results. Very rigorous attention was paid to the training of the assessors/raters. A check to identify inaccuracies in data entry was performed. Also, statistical analysis is correct. Importantly, as recommended in the STARD statement raw counts were included and comments with the open-ended item on circumstances for any adverse events were evaluated.
The results are presented in a very clear manner. I appreciate very much the cross tabulations and the graphical representation of flow chart of inclusion of residents. A remarkable feature of this tool emerging from the study is that the correlation of the two PAIC15 assessments with the pain instruments used as pain standards was highest for the PAINAD and lowest for self-report, with ROC discrimination consistently good for both assessments against the cut off for the PAINAD only.
As explained in detail in the Results section, the Authors conclude in the Discussion that the cut off for possible pain on the PAIC15 is 3 when self-report is prioritized over structured observation and an observer estimate of pain and sensitivity is prioritized over specificity and of 4 for probable pain. Based on the obtained results, the discussion highlights the risk of underestimating pain with observations compared to self-report and that the knowledge of the patients by the observer helps in pain assessment (eliciting self-reported pain or better estimates). Anothr interesting result of this study is that three quarters of the nursing home patients were found to be able to self-report pain on a combined numerical/verbal/colour scale in agreement with literature. I do agree with the observation of the Authors that situations in which pain is being induced (e.g. during care activities or as part of the pain assessment) can influence the cut off values and that, therfore, future studies determining cut offs in these circumstances are needed.
I appreciate it very much that in the Conclusions the Authors provide the reader with a precise recommendation on the choice of the cut off of 3 or 4 depending on the context (clinical practice or research).
Minor points:
The abbreviation of Global Deterioration Scale (GDS) should be firstly reported in the 2.2 Data Collection section (please, see line 113).
There is a typo (6. Patents) at line 437.
Author Response
refer to uploaded response

Reviewer 3 Report
The authors report their assessment of the Pain Assessment in Impaired Cognition (PAIC15) instrument in patients with dementia. They compare PAIC15 to patient self report, the PAINAD scale, and observers estimate. They determine that a score of 3 or greater on the PAIC15 is associated with probable pain.
The manuscript is a bit wordy. With appropriate editing, it could be condensed. The second paragraph of the conclusion should probably be moved to the discussion, as it is an interpretation of the results, not a conclusion based on the results. The authors should justify why they did not chose to compare the PAIC15 to another validated scale of the 28 available scales to which they refer. Furthermore, they should explain why they chose the PAINAD scale, which they state does not have a cut-off value. Lastly, the authors should explain the process for the including only 137 patients in the second pain assessment.
Author Response
refer to uploaded response
